# A scaling investigation of urban form features in Latin America cities

Aureliano S. S. Paiva[1]*, Gervásio F. Santos[1,2], Caio P. Castro[1,3], Daniel A. Rodriguez[4], Usama Bilal[5,6], J. Firmino de Sousa Filho[1,2], Anderson Freitas[1], Felipe Montes[7], Iryna Dronova[8], Maurício L. Barreto[1], Roberto F. S. Andrade[1,3]

1 Center of Data and Knowledge Integration for Health (CIDACS), Instituto Gonçalo Moniz, Fundação Oswaldo Cruz, Salvador, Bahia, Brazil, 2 Economics Faculty, Federal University of Bahia, Salvador, Bahia, Brazil, 3 Institute of Physics, Federal University of Bahia, Salvador, Bahia, Brazil, 4 Department of City and Regional Planning and Institute of Transportation Studies, University of California Berkeley, Berkeley, California, United States of America, 5 Urban Health Collaborative, Drexel Dornsife School of Public Health, Philadelphia, Pennsylvania, United States of America, 6 Department of Epidemiology and Biostatistics, Drexel Dornsife School of Public Health, Philadelphia, Pennsylvania, United States of America, 7 Department of Industrial Engineering, Universidad de los Andes, Social and Health Complexity Center, Bogotá, Colombia, 8 Department of Landscape Architecture & Environmental Planning, University of California Berkeley, Berkeley, California, United States of America

* sanchobuendia@gmail.com

**Data Availability Statement:** "Availability of data and materials The code and implementation of scale analysis is available on Github. https://github.com/sanchobuendia/Scaling-urban-form The

## Abstract

This paper examines scaling behaviors of urban landscape and street design metrics with respect to city population in Latin America. We used data from the SALURBAL project, which has compiled and harmonized data on health, social, and built environment for 371 Latin American cities above 100,000 inhabitants. These metrics included total urbanized area, effective mesh size, area in km$^2$ and number of streets. We obtained scaling relations by regressing log(metric) on log (city population). The results show an overall sub-linear scaling behavior of most variables, indicating a relatively lower value of each variable in larger cities. We also explored the potential influence of colonization on the current built environment, by analyzing cities colonized by Portuguese (Brazilian cities) or Spaniards (Other cities in Latin America) separately. We found that the scaling behaviors are similar for both sets of cities.

## Introduction

The growth of cities is characterized by many features that evolve over time [1], resulting from the action of many events and actors. Individual behaviors [2, 3], social institutions, economic interests, available technologies, and cultural tracts and values leave their signatures in each city. Despite the specificities of these actors in each city, such driving forces succeed in providing over time quite similar features across cities. For instance, cities are resilient, exhibit inequality patterns generated by intense competition for space, resources, and transportation [4–8], require the input of energy for their maintenance, and lack balance during their growth [9–12]. A series of prior studies have identified that many measurable features of cities, covering socio-economic, infrastructure, and health aspects, are found to depend on their size

SALURBAL project welcomes queries from anyone interested in learning more about its dataset and potential access to data. To learn more about SALURBAL's dataset, visit https://drexel.edu/lac/ and the data can be requested on the Salurbal website at the following address https://drexel.edu/lac/about/contact/."

**Funding:** The Salud Urbana en América Latina (SALURBAL)/Urban Health in Latin America project was funded by the Wellcome Trust [205177/Z/16/Z]. CIDACS has support from the Wellcome Trust UK (202912/B/16/Z) Instituto Gonçalo Moniz, Fundação Oswaldo Cruz, Salvador, Bahia, Brazil. UB was also supported by Office of the Director of the National Institutes of Health under award number DP5OD026429. The funders had no role in study design, data collection and analysis, decision to publish, or preparation of the manuscript.

**Competing interests:** The authors have declared that no competing interests exist.

according to power-laws with respect to the city size [13–19]. Such simple mathematical laws express the presence of scaling behavior, a remarkable property of complex systems, which supports identifying and studying cities within such a general framework.

The identification of scaling behaviors has also received support from a robust theoretical framework that explains how scaling relations between social-economic indicators and city size emerge as a consequence of universal aspects of the drivers of city growth, which do not depend on particularities of the actors or their past history [1, 13–14, 18, 19]. The foundational assumptions of scaling theory include the importance of human interactions in physical space, the role of social networks in enhancing individual productivity and quality, the notion that all human activities generate costs and benefits, and the bounded nature of human effort. Moreover, scaling theory considers that the size of human settlements is both a consequence and a determinant of productivity, technological and cultural development, highlighting the importance of distributed knowledge and social networks in shaping economic and social outcomes. By recognizing the role of social networks in promoting knowledge sharing and innovation, it provides a better understanding of how to promote economic and social development in urban areas. In addition, a complementary aspect to the more usual cross-sectional scaling analysis, based on the analysis of city measures for a fixed value of time, refers to the temporal scaling properties. In this case, the focus lies on how these same measures scale with respect to time for individual cities. By comparing their evolution, one may further disclose how particularities in city growth interfere in their time evolution. This topic has called the attention of several authors more recently [2, 20, 21].

In general, city growth positively increases social and economic returns, which is expressed in terms of super-linear scaling relations between the indicator and city size. On the other hand, per-capita expenditures necessary to set up adequate infrastructure in roads, electric cables, school, and health care institutions decrease with city size, resulting in sub-linear scaling relations [22, 23]. Finally, the scaling framework also reveals usual linear relations (i.e., similar per capita values) of indicators related to needs of individuals, such as number of jobs, water, and electricity consumption [19, 24–26].

Knowing the scaling patterns of city size and other indicators is important for urban planning and policy-making. Given the current pace of the worldwide urbanization phenomenon, scaling analysis provides valuable insights on how water and sewer infrastructure, urban transport, and even individual health may vary with changing population. Planning should also aim at reducing the scaling up of undesirable consequences of city growth, including high levels of pollution, congestion, inequality and segregation across city subareas, warranting accessible education and health care, providing healthy environment, and preventing natural disasters [15–17, 22, 24, 27].

In this work we are mainly concerned with identifying and characterizing physical characteristics of cities that are usually grouped under the umbrella term "urban form". In the huge literature on city scaling properties, less attention has been devoted to these specific traits. They include size, materials, street characteristics, configuration, and spatial arrangement of the built and natural environment, as well as the transport networks [9]. Once the urban form indicators result from the same factors driving city growth that leads to optimized efficiency in economic performance and infrastructure, they are expected to show similar scaling relations, despite the intricate relationships among different aspects of its dynamics.

We consider two additional topics to deepen our perspective on scaling behaviors. We note that the number of studies based on data from Latin America cities in countries formerly colonized by Spaniards or Portuguese is significantly smaller than those in countries colonized by the British or with older cities. Thus, our investigation also considers whether scaling analysis can detect a possible influence of the early phase of city growth in the region. Cities colonized

by Spaniards were mostly grounded inland, on carefully chosen and strategic plateau sites, according to geometric grid designs. However, Brazilian cities grew more organically, spreading along coastal area and less constrained by administrative blueprints of city form [28]. The contrast between the Brazilian and Spanish American cities is symbolized by their physical plans [25]. While a Spanish American city is composed of a *plaza mayor* surrounded by gridded streets, the typical Portuguese city tradition does not include a *plaza mayor*. A yet unknown issue is whether and how such early phase differences have contributed to distinct scaling behaviors when the two sets of cities are analyzed jointly and separately. Results for all cities in the SALURBAL data set have also been compared with those provided by two separate subsets, the first one consisting of cities in Brazil (colonized by Portugal), and the other including cities in regions of Latin American that were colonized by other countries (mainly by Spain). In this case, a temporal scaling analysis [2, 20, 21] for some representative cities colonized Portugal and Spain might bring complementary information to the results we discuss here. This issue, however, will be postponed to another investigation, due to the lack of long-term longitudinal data on urban form for these 371 cities.

In this work we used harmonized data from the Urban Health in Latin America Project (Salud Urbana en America Latina, or SALURBAL), a partnership of the Drexel University Dornsife School of Public Health and 14 institutions across Latin America and the United States. For this analysis, the SALURBAL data are available at three urban levels: the metropolitan area; the urban extent, corresponding to the urban footprint regardless of administrative boundaries; and the component units of metropolitan areas, such as municipalities. Thus, the data allows characterizing the scaling behavior of built environment variables and analyze whether results are sensitive to boundary definitions.

More and more evidence indicate that scaling behavior prevails in a much clear way when cities are defined as social networks embedded in physical space, as compared to results using city definitions based on administrative and political entities [29]. Despite that, we take advantage of the available SALURBAL data and perform scaling analysis for the three levels of city definition as defined above. As we will discuss later, our results provide support to a much clear scaling behavior when the urban extent city definition is used. We selected and worked with the following urban form measures: *Total urban area, Number of urban patches, Effective mesh size, Total administrative area (or Area in km$^2$ in the Salurbal dataset), Large roads, Intersection, Intersection 3-way, Intersection 4-way* and *Number of streets* [26, 30–32]. In the (Data/Methods) section we provide a detailed description of these variables.

## Methods

Power-laws describe a functional relationship between any two sets of measured quantities (say $Y_i$ and $N_i$) that can be expressed as

$$Y_i = Y_0 N_i^{\beta}, \tag{1}$$

such as the area and perimeter of a square, circle or other simple geometrical forms, (where $Y_i$ ↔ area, $N_i$ ↔ perimeter or other characteristic length, $\beta = 2$, and $Y_0$ is a constant depending on the geometrical form). The scaling framework is based on the identification that Eq (1) is valid given two variables characterizing the system under scrutiny, where now $\beta$, not necessarily an integer number, is called the scaling exponent that characterizes the dependency between $Y$ and $N$, and $Y_0$ is a constant depending on the used measure unit. In this study, $Y$ represents a city urban form characteristic, as discussed in the sequence, $N$ is the population size, and $Y_0$ is a constant related to the typical magnitude of that measure in the sample. The subscript $i \in [1, n]$, where n indicates the number of cities. $\beta$ determines the relative increase

rate of $Y$ in terms of $N$. If we take the logarithm on both sides of Eq (1) we obtain

$$log(Y_i) = log(Y_0) + \beta log(N_i). \qquad (2)$$

We worked with logarithms of the variables in base 10, while $log(Y_0)$ and $\beta$ were estimated by ordinary least squares regression, where $\beta$ is the slope. Besides assessing the goodness of fit by the usual parameter $R^2$, the distribution of residuals between the actual points and the best fitting line was also evaluated. The results provided by Eq (2) can be used to make predictions. If we know the population of a city that was not part of the estimation sample, its expected value of $Y$ can be estimated based on its population.

Table 1 summarizes the interpretation of scale exponents according to mathematical definitions: $\beta = 1$ indicates a linear isometric relation between $Y$ and $N$ indicating that the relative value of the variable does not vary with respect to the city size. When $\beta > 1$ or $\beta < 1$, associations are labeled as superlinear ($Y$ increases faster than population) and sublinear ($Y$ increases slower than population), respectively [10–12, 23, 32–34]. All analyses were conducted using the Python programming language.

Besides the information on city population *(N)*, nine urban form related measures from the SALURBAL data set were used in the scaling analysis (see Ref. [35] for a detailed description of the used data set). Summarized definitions are as follows:

(i) *N*, representing the total population within the indicated administrative boundary, was adjusted for United Nations' country-level population projections. (ii) *Total urban area* is defined as the total urbanized area in the city, based on the count of the number of 30m x 30m grid-cells classified as urbanized in the geographic boundary. (iii) An urban patch is defined as a contiguous region classified as urbanized, and the *Number of urban patches* is just its number within a geographic boundary. (iv) *Effective mesh size* is defined as the sum of squares of urban patch size, divided by the total area of the geographic unit. It is an indicator of fragmentation; whereby larger patches have higher weights. Larger values reflect a larger size of urban development; low values reflect scattered, small development. These observations suggest that this urban form variable can behave superlinearly. (v) *Total administrative area* corresponds to its official value as provided by each country. (vi) *Large road* measures the total length of streets with three or more lanes within the geographic unit. It was obtained by multiplying the *large road density* values at the SALURBAL data set by the *Total urban area*. The same strategy was adopted for the next four urban form measures. (vii) *Intersection* measures the number of street intersections within the geographic unit. (viii) *Intersection 3* measures the number of street intersections where exactly 3 streets meet within the geographic unit. (ix) *Intersection 4* measures the number of intersections where exactly 4 streets join within the geographic unit. (x) *Streets* measures the total length of streets within the geographic unit.

The SALURBAL study protocol was approved by the Drexel University Institutional Review Board with ID #1612005035 and by appropriate site-specific IRBs.

**Table 1. Classification of scaling exponents for urban properties.**

| Scaling exponent | Classification | Definition |
|---|---|---|
| $\beta < 1$ | Sublinear | Relatively lower values in large population cities |
| $\beta = 1$ | Linear | Similar values across different city with any population |
| $\beta > 1$ | Superlinear | Relatively higher values in large population cities |

## Data

We use data from the SALURBAL Project, which has compiled and harmonized data on health, and the social and built environment for Latin American cities, comprising nearly 300 million people. The project includes cities in 11 countries: Argentina, Brazil, Chile, Colombia, Costa Rica, El Salvador, Guatemala, Mexico, Nicaragua, Panama, and Peru. Being more specific, we restrain our study to urban agglomerations classified as "cities" at the L1 and L2 SALURBAL levels [35], which amounts to taking $n = 371$ or 1436 at the L1 or L2 levels, respectively. At L2 level, a city always corresponds to its political/administrative definition. However, at L1 level we make use of two different agglomeration criteria for "city" definition: L1Admin (or L1AD) corresponds either to an administrative L2 unit or a combination of adjacent L2 units with continuous urbanization; in L1UrbExt (L1UX), a "city" follows from a systematic identification of continuous urban extent of built area obtained from satellite imagery.

For the sake of achieving a fair harmonization of data from widespread origins, the SALURBAL L1 data set only comprises cities with 100,000 residents or more. While there is no universal definition of city, larger population sizes are necessary to generate a critical mass of residents. Fig 1 shows the spatial location of L1 cities, where Brazil has the largest share of cities in a single country (152) followed by Mexico (92).

## Results

### Descriptive statistics

Based on the SALURBAL data sets indicated in the previous section, we show in Table 2 quantitative information on the number of cities of each country and its corresponding percentage with respect to the total number of cities in the data set. It also presents a summary of the data main features, i.e., the mean values at country level of the 9 used urban form variables and population size using the L1AD city definition: *Total population, Total urban area, Number of urban patches, Effective mesh size, Total administrative area, large road, Intersection, Intersection 3-way, Intersection 4-way, Street*. Entries in the last five columns correspond to the original street metrics densities in the SALURBAL data set (i.e., actual city number of occurrence divided by city area), from which we calculated the actual city values used in our analyses as described in the Methods section.

To providing a simple but otherwise important comparison between the contributions coming from cities in Brazil (BR) and from the other Latin American countries (OtherLA), for all variables listed in Table 2 we evaluated the average values restricted to the cities in the BR and the OtherLA subsets and their corresponding ratio, which we indicate as *BR/OtherLA*. Table 3 shows the obtained results using the three city levels definitions. As the values in the last five columns of Table 2 reproduce the original SALURBAL values of street metrics of each city in terms of their densities, from which we obtained their actual value, we decided to include in Table 3, whenever pertinent, the corresponding values of the variable *BR/OtherLA* based on the density values.

Starting with the L2 city definition we see that, for the *Total population, Total urban area, Number of urban patches* and *Effective mesh size* variables, the ratio *BR/OtherLA* stay in the interval (0.96,1.10), indicating that these variables have similar values in both subsets. On the other hand, for the *large road, Intersection, Intersection 3-way* and *Street* variables, we have $1.08 < BR/OtherLA < 1.35$. These larger values to Brazilian cities for these ratios, together with the 0.87 ratio for *Intersection 4-way*, can indicate a more disorganized structure in Brazilian cities. Indeed, the number of *Intersection 4-way* occurrences captures, quite directly, the presence of a more organized structure and better division of the urban patches. Therefore, the

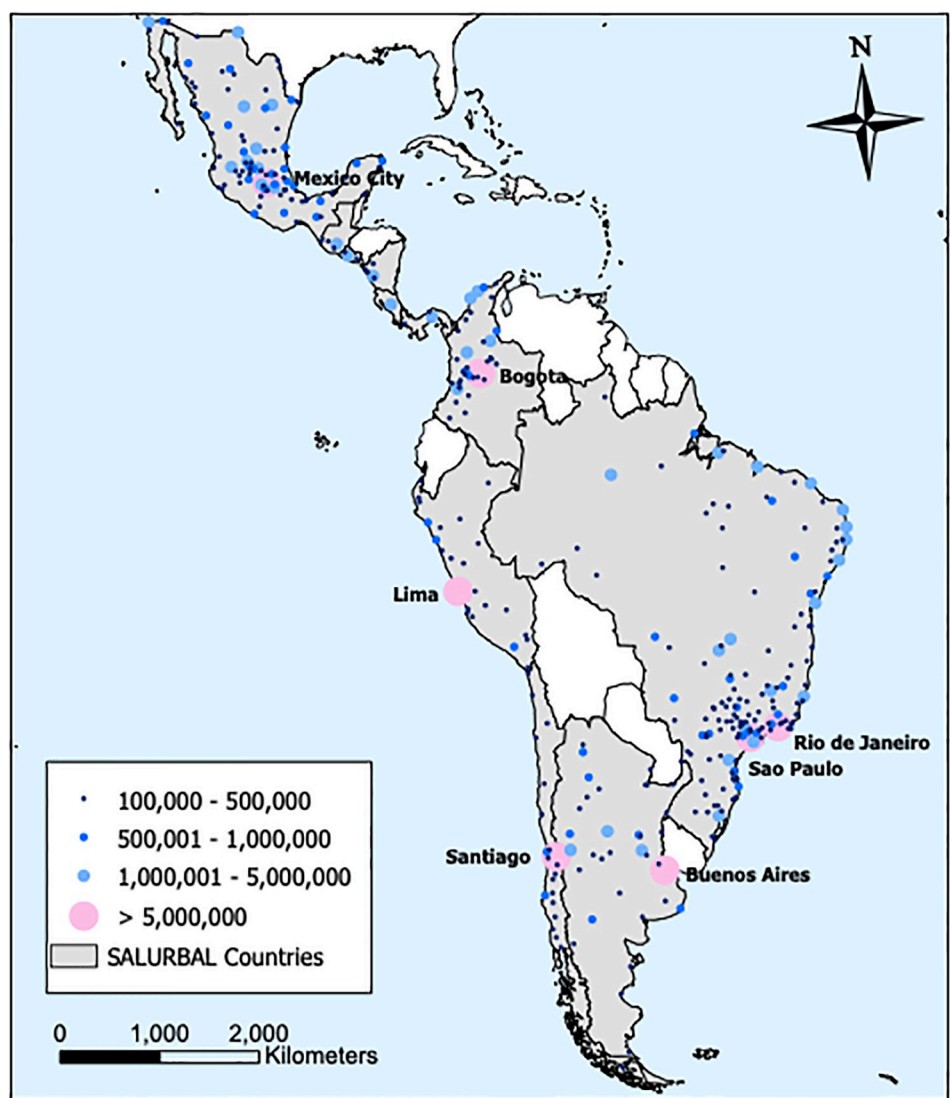

**Fig 1. Latin America map\*.** Spatial distribution and size distribution of SALURBAL cities. * Map built using country boundaries from NaturalEarthdata.com and the city locations were created by the SALURBAL Built Environment Core.

value in Table 3 stays in accordance with the hypothesis that Brazilian cities differ from other cities in Latin America in terms of the organization of their urban mesh.

The values of *BR/OtherLA* for the L1AD city definition are, generally, larger than those for the L2. This behavior can be explained by the difference between cities in the metropolitan regions, where the main city is usually much larger than the others. For *Large Road* and *Intersection 4-way*, results stay ~ 1, but for all other variables we observe that $1.35 < BR/OtherLA < 1.79$, a clear indication that all of them take larger values for cities in the BR subset as compared to cities in the OtherLA.

Table 3 shows that the L1UX city definition, based purely on satellite imagery, highlights other aspects related to the difference between the groups. The largest differences between cities of the two subsets occur in *Number of urban patches*, *Effective mesh size*, *large road* and

**Table 2. Average values of the analyzed L1AD variables over the cities in each country expressed as counts, hectare, and km.**

| Countries | Number of cities | % | Mean values by country | | | | | | | | |
|---|---|---|---|---|---|---|---|---|---|---|---|
| | | | Population | Urban area (hectare) | Number of urban patches | Effective mesh size (hectare) | Length of large roads (km) | Number of Intersections | Number of Intersections3 | Number of Intersections4 | Sum of length of streets (km) |
| BR | 152 | 40.9 | 607859 | 8184 | 207 | 3181 | 55.8 | 71522 | 51402 | 19497 | 12108 |
| MX | 92 | 24.8 | 641057 | 9150 | 157 | 4601 | 78.3 | 83921 | 56886 | 26295 | 13763 |
| CO | 35 | 9.43 | 670077 | 3661 | 62 | 1899 | 31.3 | 48952 | 33248 | 14994 | 6665 |
| AR | 33 | 8.89 | 750784 | 11719 | 240 | 5523 | 143.7 | 86657 | 44356 | 41272 | 15744 |
| PE | 23 | 6.2 | 651206 | 4849 | 65 | 2702 | 59.4 | 67598 | 4743 | 19759 | 9349 |
| CL | 21 | 5.66 | 484398 | 5695 | 85 | 3101 | 65.2 | 67917 | 51327 | 16134 | 8880 |
| NI | 5 | 1.35 | 275795 | 2851 | 99 | 1499 | 56.8 | 28713 | 20269 | 8215 | 4276 |
| SV | 3 | 0.81 | 661411 | 5449 | 212 | 2159 | 122.6 | 48649 | 37161 | 11181 | 7831 |
| PA | 3 | 0.81 | 463331 | 6483 | 186 | 3126 | 128.8 | 41573 | 32207 | 8966 | 8284 |
| GT | 3 | 0.81 | 924233 | 8035 | 246 | 2607 | 92.4 | 82903 | 64196 | 17869 | 12362 |
| CR | 1 | 0.27 | 1867603 | 27118 | 915 | 12070 | 540.7 | 173815 | 142561 | 29353 | 31620 |
| Total | 371 | 100 | 631116 | 7906 | 169 | 3574 | 70.4 | 71521 | 51402 | 19497 | 11858 |

*Intersection 4-way*. *Number of urban patches* is 45% larger for Brazilian cities, meaning that they are more fragmented. At the same time the average size of their patches and streets are smaller. In accordance with the results based on the L1AD city definition, the more fragmented Brazilian cities also have a smaller number of *Intersection 4-way*.

## Scaling analysis

A thorough search in the literature on scaling laws relating urban variables and population size revealed that most recent studies were focused on time spent during daily travel [2, 5], mass public transport [6], air pollution emissions from vehicles [7], transport networks [8]. Results for landscape metrics and street design metrics.

Here we first succeeded in identifying the presence of scaling laws for the landscape metrics variables presented in Table 4 using three different city level definitions. The variables *Total urban area*, *Number of urban patches* and *Area in km²* were found to have a sublinear

**Table 3. Ratio of values for *BR over Other LA* cities for all used variables according to three city definitions.**

| Variables | L1AD | L2 | L1UX |
|---|---|---|---|
| *Total population* | 1.5758 | 0.9449 | 0.9391 |
| *Total urban area* | 1.7564 | 1.0531 | 1.0611 |
| *Number of urban patches* | 1.7837 | 1.0904 | 1.4505 |
| *Effective mesh size* | 1.3590 | 0.9682 | 0.8269 |
| *Large road* | 1.0286 | 1.0880 | 0.6928 |
| *Intersection* | 1.4570 | 1.1937 | 0.9722 |
| *Intersection 3-way* | 1.6584 | 1.3463 | 1.054 |
| *Intersection 4-way* | 1.0577 | 0.8694 | 0.8081 |
| *Street* | 1.5894 | 1.2848 | 1.0362 |

L1AD are metropolitan areas defined by administrative units, L1UX are metropolitan areas defined by satellite imagery, and L2 are the administrative units (municipalities) that define L1ADs.

**Table 4. Landscape metrics—Values of the exponent β, confidence interval *CI*, and $R^2$ for three SALURBAL city definitions: L1 Administrative units (L1AD), L1 Urban Extent (L1UX) and L2 Administrative units (L2).**

| Variables | L1AD | | | L1UX | | | L2 | | |
|---|---|---|---|---|---|---|---|---|---|
| | $\beta_{MA}$ | 95%CI | $R^2$ (%) | $\beta_{UX}$ | 95%CI | $R^2$ (%) | $\beta_{AD}$ | 95%CI | $R^2$ (%) |
| Total urban area | 0.908 | (0.871, 0.945) | 81.8 | 0.889 | (0.855, 0.924) | 82.9 | 0.960 | (0.942, 0.978) | 84.2 |
| Number of urban patches | 0.670 | (0.619, 0.720) | 56.6 | 0.755 | (0.707, 0.802) | 65.1 | 0.451 | (0.406, 0.496) | 15.8 |
| Effective mesh size | 1.503 | (1.401, 1.606) | 61.3 | 0.909 | (0.856, 0.963) | 68.2 | 1.494 | (1.432, 1.556) | 52.2 |
| Total administrative area (or Area in $km^2$) | 0.312 | (0.223, 0.402) | 8.30 | 0.849 | (0.81, 0.888) | 78.1 | 0.572 | (0.521, 0.624) | 19.0 |

L1AD are metropolitan areas defined by administrative units, L1UX are metropolitan areas defined by satellite imagery, and L2 are the administrative units (municipalities) that define L1ADs.

dependence, while the variable *Effective mesh size*, which can be understood as a measure of granularity of city blocks, may have a super-linear behavior. These results indicate heterogeneity in the scaling relationships.

The values of R2 in Table 4, which ranges from 0 to 100%, indicate how well the data fits to the linear regression. These values are high for most of the variables and city definitions. However, those for Total Administrative area score with L1AD and L2 definitions are low, 8.30% and 19.0%, respectively. The same occurs for Number of urban patches in L2, with 15.8%. Despite of these three values, the whole set of results indicates high variability of the Number of patches and Total Administrative area of cities with similar populations if they are defined using local administrative units L2 and L1AD. The variable that better explains the increase in population and the fragmentation of cities is Number of urban patches (L2 and L1UX).

Figs 2 and 3 illustrates the data and the resulting scaling result obtained by linear regression performed on the logarithms of the two used variables. The points representing the cities are divided into two groups, according to whether they belong to the BR (green) or Other LA (orange) subsets. The straight line corresponds to the regression result when all cities are taken into account.

As consistently reported in the literature, urban metrics generally are likely to have a sublinear scaling behavior. Our results stay in line with this trend: With the L1UX city definition, all variables present a sublinear behavior, as exemplified in Fig 3 for the *Area* variable. The L2 and L1AD city definitions lead to the expected results, only with a single discrepancy in form of a superlinear behavior for *Effective mesh size* shown in Fig 2.

The results for the street design metrics in Table 5 also presented the high values of R2, except for the variable *large road* of all administrative areas, and for the variables Intersection 4-way and number of streets of cities with administrative areas of L2. Despite of this, the statistical significance of parameters is high. The *large road* variable has an exponent β very close to 1 in the L1AD and L1UX levels. Since both confidence intervals include values larger than 1, we cannot rule out a linear behavior.

The regressions for the variables *Intersection*, *Intersection 3-way* and *Intersection 4-way* consistently indicate β < 1 for all city definitions and upper limits of the confidence smaller than 1. Figs 4 and 5 illustrate with data points and resulting regression line the sublinear behavior for *Intersection 3-way* and *Intersection 4-way* for the city definitions L1AD and L2, respectively. Figs 6 and 7 show the residuals distributions for the fittings in Figs 4 and 6. In Fig 6, the histogram indicates a left-skewed distribution of residuals, while on Fig 7 the residuals distribution is more symmetric, so that it can be better approximated by the normal distribution.

After using a well curated data set that includes cities in a less studied region, our results indicate that, except for super-linear scaling behavior for the *Effective Mesh Size* variable using

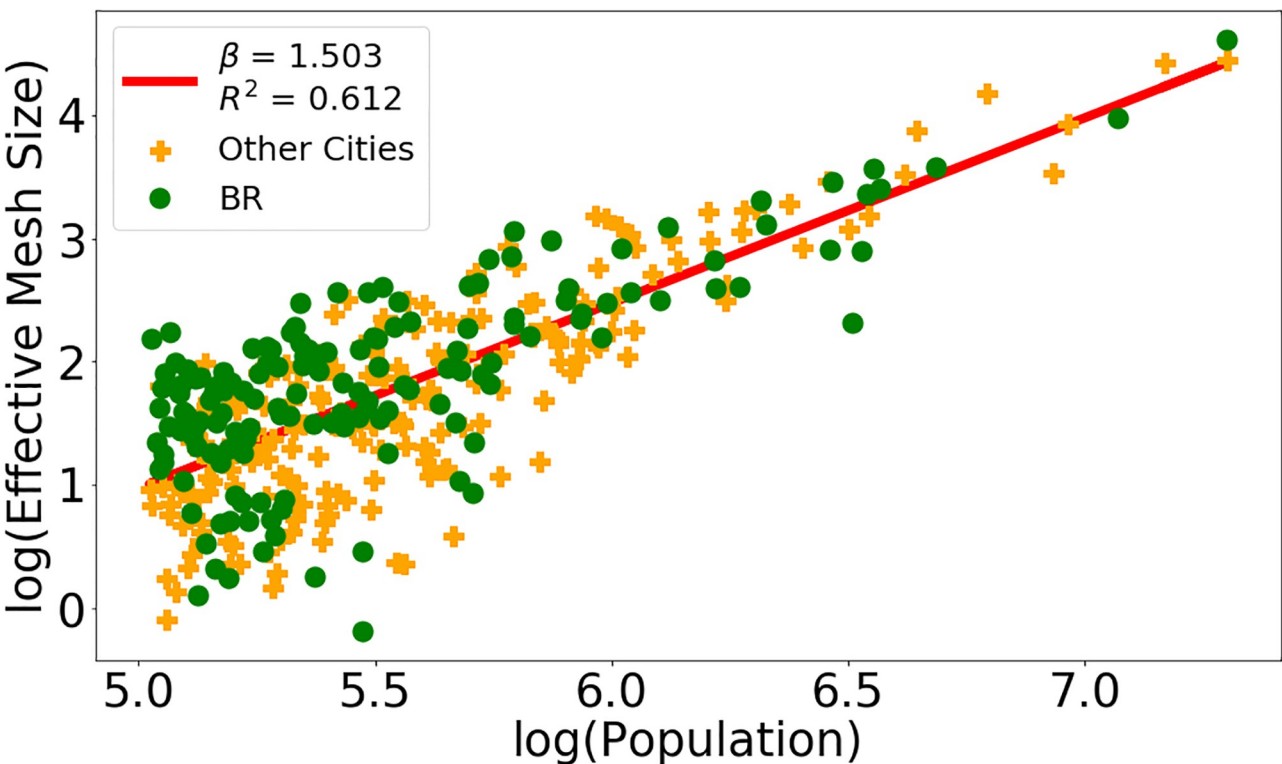

**Fig 2. Effective mesh size and area in km².** Panel (a): Data and straight line resulting from linear regression on the logarithms of the *Effective Mesh Size* and *Population* variables using the L2 city definition. Panel (b): Same contents for *Area in km²* and *Population* with the L1UX.

the L2 and L1AD city definition, the scaling properties of urban form variables consistently show a sublinear behavior that do not depend on the definition of urban boundaries (L2, L1AD or L1UX). These findings provide further support and expand the understanding on the relationship between urban metrics and the size of the population. The data set also enabled us to explore whether this general trend shows any sort of regional dependence, by splitting the analysis to include cities located in Brazil, colonized by Portugal, or in the other LA countries of SALURBAL, mainly colonized by Spain. Despite the fact, revealed by the descriptive analysis of the metric variables and by satellite imagery from above, that cities colonized by Spaniards have a more organized fragmentation of the blocks than those colonized by Portugal, we found no significant differences in the exponents when the analyses are carried out with cities in either BR or Other LA subsets for any of the other city definitions (see Table 6). This includes the super-linear behavior for the *Effective Mesh Size* variable based on L2 and L1AD city definitions.

## Discussion

This study presents the first-time analysis of urban scaling patterns for the built environment of Latin American cities using three different city definitions. The results reveal that sublinear behavior is dominant for built environment variables with respect to population size. The sublinear exponents associated to infrastructure networks found in [3] using a fractal geometry perspective, which stays in line with general grounds discussed in [19], reveal that our results consolidate the understanding that street metrics follow a sublinear behavior. This general

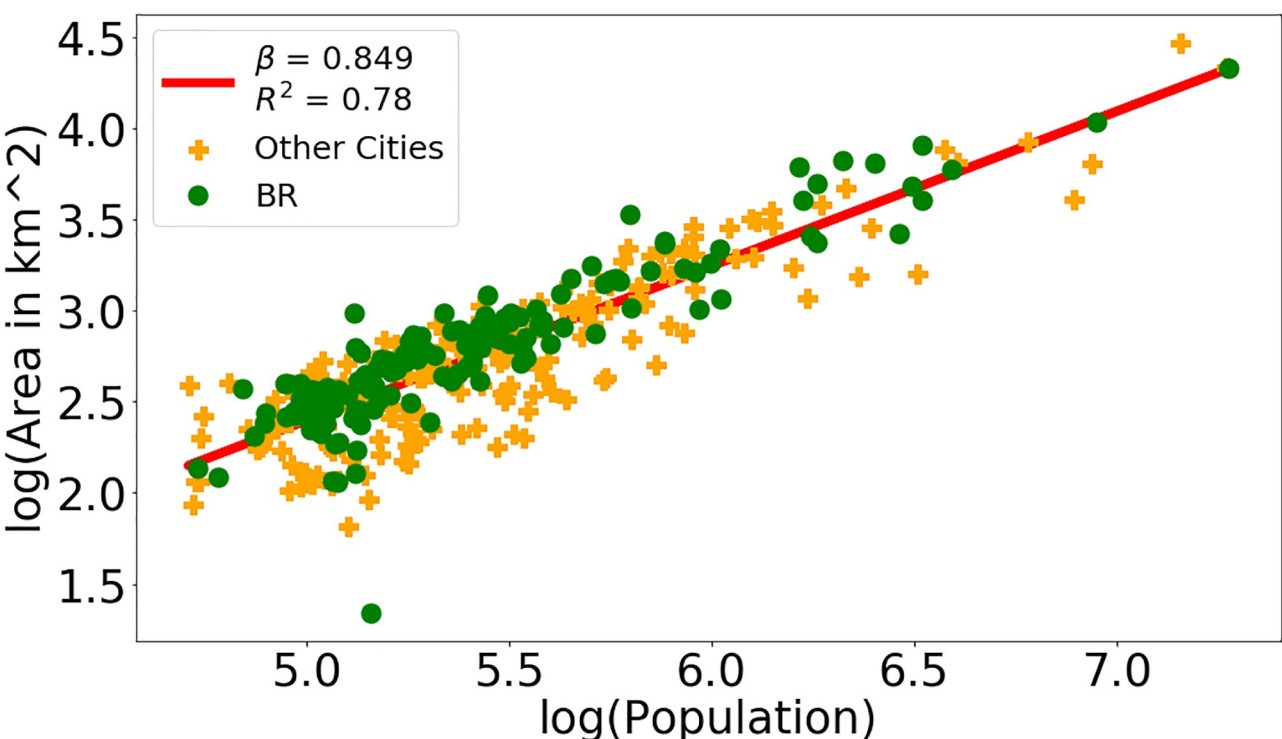

**Fig 3. Effective mesh size and area in km².** Panel (a): Data and straight line resulting from linear regression on the logarithms of the *Effective Mesh Size* and *Population* variables using the L2 city definition. Panel (b): Same contents for *Area in km²* and *Population* with the L1UX.

trend reproduced our results can be put in perspective with the efficiency cities have in absorbing population within the urban structure as they grow.

The $\beta$ values found for *Total Urban Area* in L2 and L1UX city definitions are close of those found in [36], both being sublinear. Also, the sublinear behavior found here for most variables are in agreement with previous findings for other world regions [19].

Analyzing temporal data, studies [37] found that the number of nodes $N$ of intersections is a linear function of the number of people living in a certain city. Considering the noise in data and that our database encompasses many cities, the results reported here are in relative agreement with their results.

**Table 5. Street design metrics—Values of the exponent $\beta$, confidence interval *CI*, and $R^2$ for three SALURBAL city definitions.**

| Variables | L1AD | | | L1UX | | | L2 | | |
|---|---|---|---|---|---|---|---|---|---|
| | $\beta_{MA}$ | 95%CI | $R^2$(%) | $\beta_{UX}$ | 95%CI | $R^2$ (%) | $\beta_{AD}$ | 95%CI | $R^2$ (%) |
| *Large road* | 0.927 | (0.757,1.098) | 28.8 | 0.999 | (0.846,1.152) | 38.0 | 0.520 | (0.421, 0.618) | 12.6 |
| *Intersection* | 0.864 | (0.826, 0.902) | 79.1 | 0.881 | (0.846,0.916) | 82.0 | 0.954 | (0.932,0.976) | 77.5 |
| *Intersection 3-way* | 0.888 | (0.850, 0.925) | 80.2 | 0.908 | (0.874,0.943) | 83.5 | 0.923 | (0.898,0.948) | 72.4 |
| *Intersection 4-way* | 0.813 | (0.761, 0.865) | 64.0 | 0.828 | (0.777,0.879) | 66.0 | 0.935 | (0.901,0.969) | 59.0 |
| *Number of streets* | 0.753 | (0.708, 0.799) | 67.1 | 0.880 | (0.844,0.916) | 81.6 | 0.830 | (0.806,0.853) | 40.6 |

L1AD are metropolitan areas defined by administrative units, L1UX are metropolitan areas defined by satellite imagery, and L2 are the administrative units (municipalities) that define L1ADs.

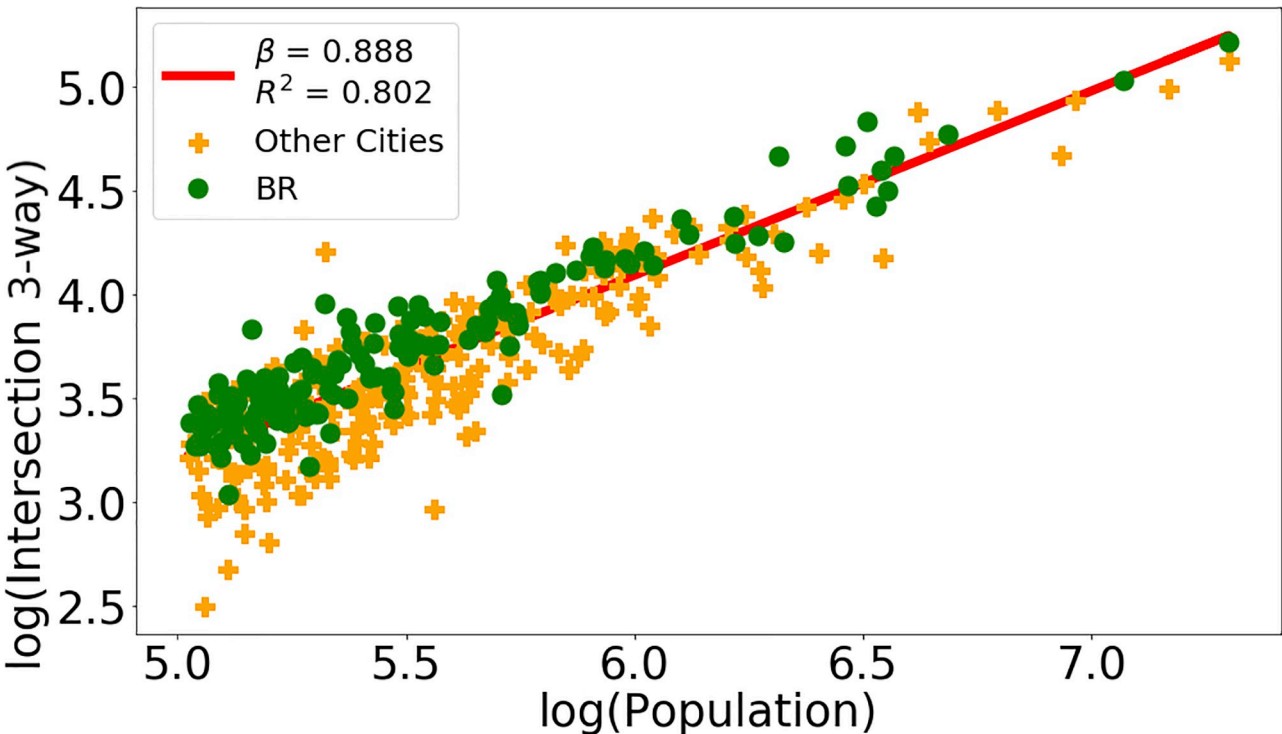

**Fig 4. Intersection 3-way and Residuals distribution.** Panel (a): Data and straight line resulting from linear regression on the logarithms of the *Intersection 3-way* and *Population* variables using the L1AD city definition. Panel (b): Residuals distribution evaluated from data and line obtained by regression shown on Panel (a).

The sublinear behavior, with low exponent, found for *Area in $km^2$* and the almost linear behavior found for *Total urban area* are in agreement with the results found for Italian cities [38], where the administrative area has a sublinear behavior with low exponent and the urban area has a linear behavior. It is particularly important to observe the second lowest values found in the *Street* variable, also indicative of cities ability of hosting larger populations with a relatively low increase in their street networks.

A second factor that needs to be explored in scaling analysis, beyond scaling coefficients, is how well the data fits the scaling model, measured using the $R^2$ metric. Here, we found that the $R^2$ was low ($< 50\%$) for some variables in different levels, indicating that the power-law scaling assumption may not be valid. Importantly, we found these low values mostly in analysis using administrative definitions (L2 and L1AD), but not when using definitions based on the actual extent of the city (L1UX), indicating that the urban extent definitions may be more appropriate to study scaling of built environment variables. A second reason that may lead to low $R^2$ is the fact that these city definitions concentrate many entries at the low end of the spectrum.

It is also important to call the attention for the super-linear behavior for the *Effective Mesh Size* variable found at the L1AD and L2 city definitions, in opposition to the sublinear behavior at the L1UX definition. This supports the above claim on the inadequacy of political/administrative city definitions in scaling studies of built environment variables.

While some average values identified in the descriptive statistics of the used variables showed some differences for the sets of Brazilian and no-Brazilian cities, our results were unable to detect any significant difference among the values of $\beta$ for the set of cities colonized

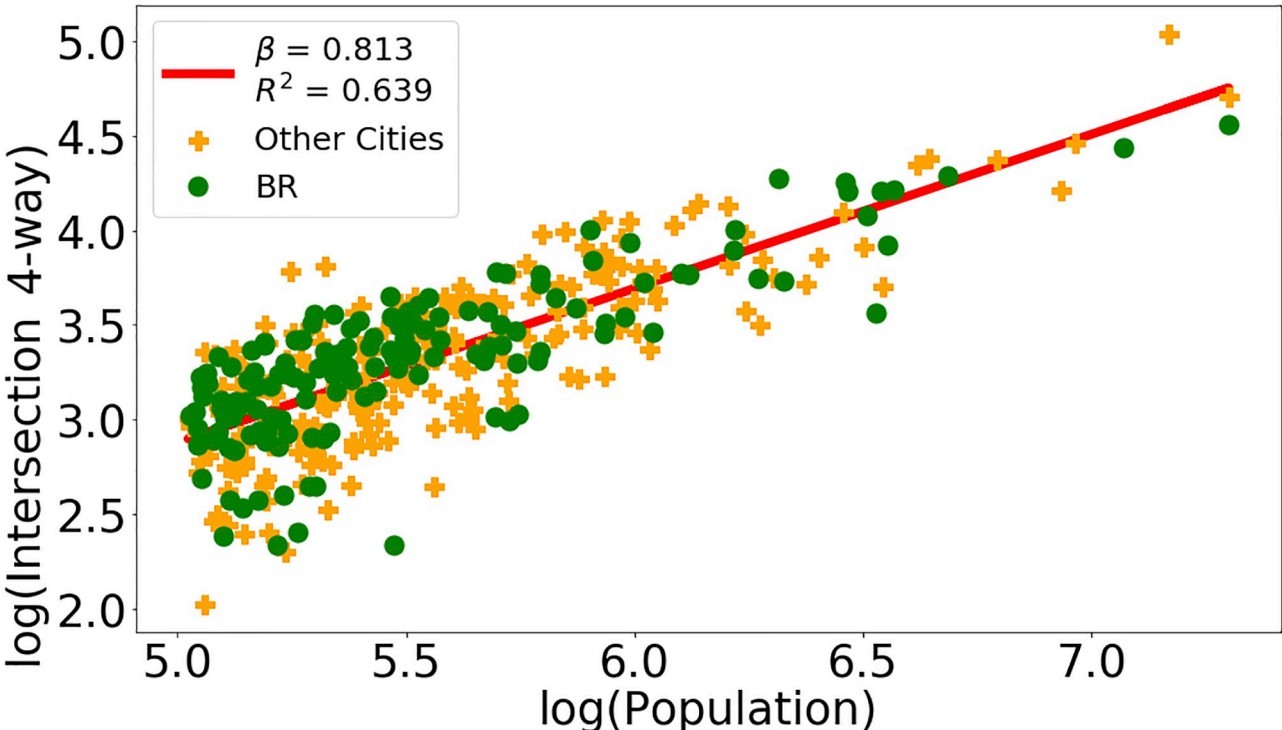

**Fig 5. Intersection 4-way and Residuals distribution.** Panel (a): Data and straight line resulting from linear regression on the logarithms of the *Intersection 4-way* and *Population* variables using the L2 city definition. Panel (b): Residuals distribution evaluated from data and line obtained by regression shown on Panel (a).

by Portuguese or Spaniards, despite previous research identifying different design patterns in sets of cities. Potential reasons for this lack of a difference include that the influence of the original colonization on the built environment, originated in the 16th and 17th centuries, has been overcome over the years by the same search of efficiency that drives city growth wherever they are.

Therefore, despite having suffered different historical influences and different ages, these cities currently share the same scaling behavior for urban variables. Alternatively, this may also mean that scaling relationships for built environment features are robust to initial differences among the cities, and that their growth reflects the universal character of the driving mechanisms responsible for keeping cities alive and growing [39]. Indeed, the results found for the scale analysis do indicate a degree of universality in the way population size is related to these variables [15–17, 22, 24, 27].

The results found here will serve as a basis for comparison with further studies targeted to similar analyses for cities in other regions. We emphasize the importance of using appropriate city definitions in scaling analyses. Our study reveals that city definitions based on the actual extent of the urban area yield more accurate insights compared to administrative definitions. This highlights the need for urban planners in Latin America to consider the physical boundaries of urban areas when making decisions about resource allocation, spatial organization, and infrastructure development. By understanding the scaling behavior of built environment variables in relation to the actual urban extent, planners can create more effective strategies to address the specific needs and challenges of rapidly expanding cities.

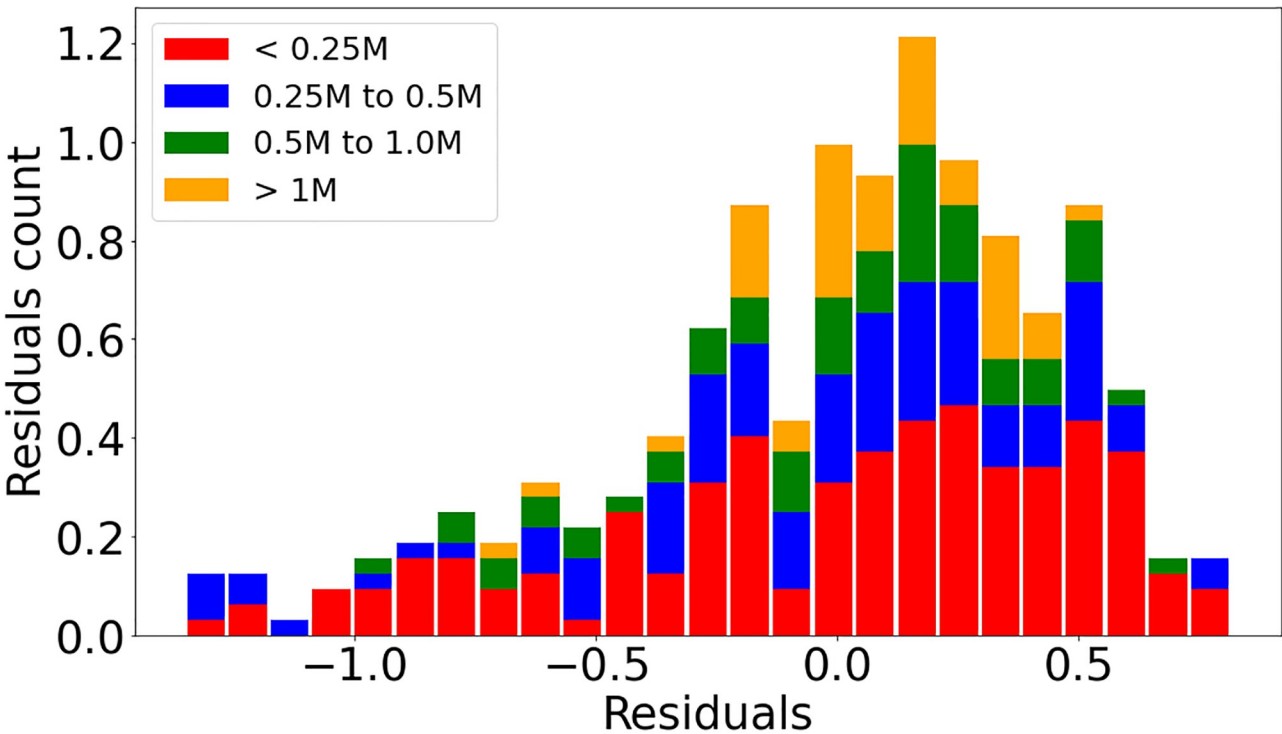

**Fig 6. Intersection 3-way and Residuals distribution.** Panel (a): Data and straight line resulting from linear regression on the logarithms of the *Intersection 3-way* and *Population* variables using the L1AD city definition. Panel (b): Residuals distribution evaluated from data and line obtained by regression shown on Panel (a).

Furthermore, the study's discovery of a degree of universality in the relationship between population size and built environment variables across diverse Latin American cities has important implications for urban planning. Despite varying economic, historical, and cultural differences, the fundamental mechanisms that drive urban growth appear to exhibit common patterns. This finding suggests that certain strategies for city development may be applicable across different cities in the region, providing valuable guidance for planners seeking sustainable and efficient urbanization approaches.

Among the limitations in scaling analysis, we must consider factors such as the amount of data and their dispersion, which influence the results and can be quantified by the residuals. Besides the low value of $R^2$ mentioned above, another feature of the data that must be considered is related to the interval of variation of the independent variable. If it is too small, which is expressed by the number of decades in a double logarithmic scale, the results may be difficult to interpret.

## Acknowledgments

The authors acknowledge the contribution of all SALURBAL project team members. For more information on SALURBAL and to see a full list of investigators see <https://drexel.edu/lac/salurbal/team/>.

SALURBAL acknowledges the contributions of many different agencies in generating, processing, facilitating access to data, or assisting with other aspects of the project. Please visit <https://drexel.edu/lac/data-evidence> for a complete list of data sources.

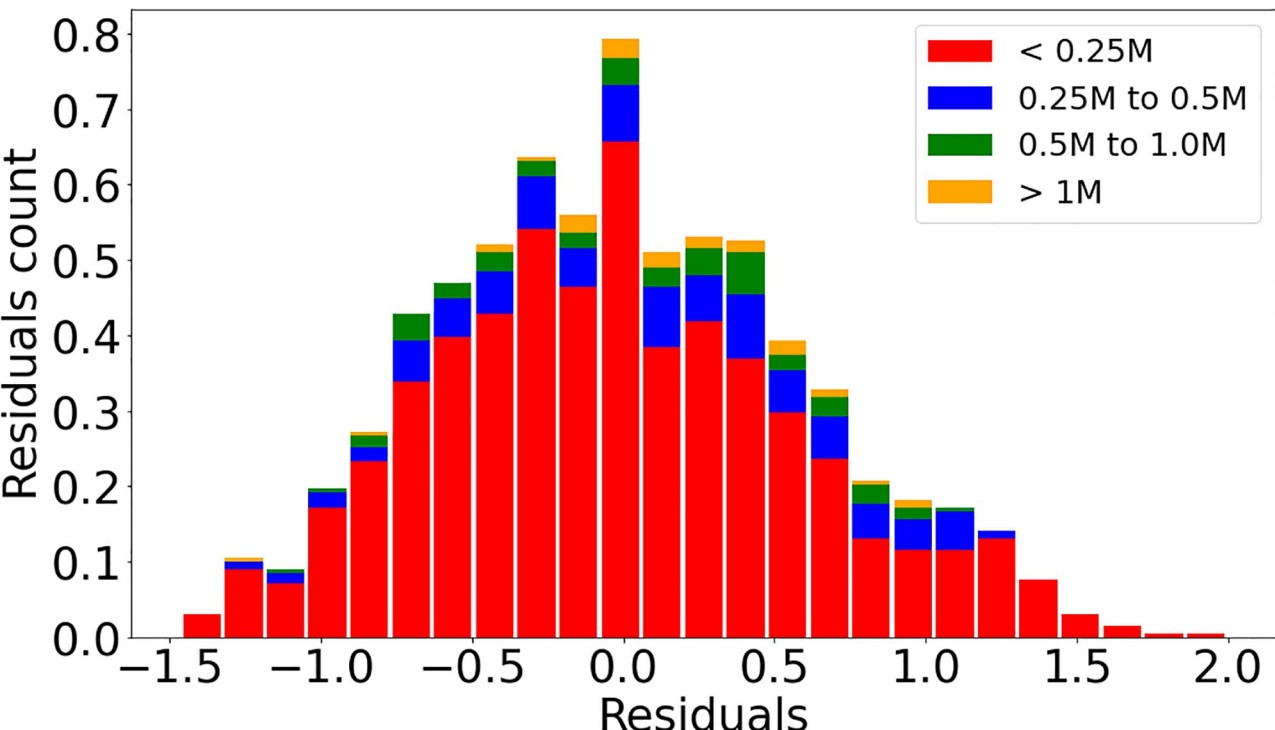

**Fig 7. Intersection 4-way and Residuals distribution.** Panel (a): Data and straight line resulting from linear regression on the logarithms of the *Intersection 4-way* and *Population* variables using the L2 city definition. Panel (b): Residuals distribution evaluated from data and line obtained by regression shown on Panel (a).

**Table 6. Landscape and street design metrics—Values of the exponent $\beta$ for three SALURBAL city definitions when cities in the two BR and LAOther subsets are analyzed separately.**

|  | L1AD | | L1UX | | L2 | |
|---|---|---|---|---|---|---|
| Variable | $\beta_{L1AD}BR$ | $\beta_{L1AD}Other$ | $\beta_{L1UX}BR$ | $\beta_{L1UX} Other$ | $\beta_{L2}BR$ | $\beta_{L2}Other$ |
| *Total urban area* | 0.899 | 0.928 | 0.891 | 0.894 | 0.864 | 0.964 |
| *Number of urban patches* | 0.725 | 0.639 | 0.822 | 0.72 | 0.394 | 0.396 |
| *Effective Mesh Size* | 1.318 | 1.675 | 0.843 | 0.953 | 1.505 | 1.51 |
| *Total administrative area (or Area in km$^2$)* | 0.378 | 0.253. | 0.866 | 0.843 | 0.413 | 0.562 |
| *Large road* | 0.727 | 0.792 | 0.862 | 0.898 | 0.668 | 0.833 |
| *Intersection* | 0.826 | 1.004 | 0.97 | 1.036 | 0.753 | 0.495 |
| *Intersection 3* | 0.828 | 0.908 | 0.859 | 0.899 | 0.785 | 0.968 |
| *Intersection 4* | 0.861 | 0.928 | 0.893 | 0.924 | 0.772 | 0.928 |
| *Number of streets* | 0.754 | 0.863 | 0.789 | 0.855 | 0.786 | 0.962 |

L1AD are metropolitan areas defined by administrative units, L1UX are metropolitan areas defined by satellite imagery, and L2 are the administrative units (municipalities) that define L1ADs. BR refers to cities in Brazil, while Other refers to cities in the 10 Other LA countries.

## Author Contributions

**Conceptualization:** Roberto F. S. Andrade.

**Data curation:** Aureliano S. S. Paiva, Usama Bilal, Anderson Freitas, Roberto F. S. Andrade.

**Formal analysis:** Aureliano S. S. Paiva, Gervásio F. Santos, Usama Bilal, Maurício L. Barreto, Roberto F. S. Andrade.

**Methodology:** Gervásio F. Santos, Usama Bilal, Maurício L. Barreto, Roberto F. S. Andrade.

**Visualization:** Roberto F. S. Andrade.

**Writing – original draft:** Aureliano S. S. Paiva, Gervásio F. Santos, Caio P. Castro, Daniel A. Rodriguez, Usama Bilal, J. Firmino de Sousa Filho, Felipe Montes, Iryna Dronova, Maurício L. Barreto, Roberto F. S. Andrade.

**Writing – review & editing:** Aureliano S. S. Paiva, Gervásio F. Santos, Usama Bilal, Maurício L. Barreto, Roberto F. S. Andrade.

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
