## [Decision Letter · Decision Letter 0]

16 Feb 2023

PONE-D-22-28772A scaling investigation of urban form features on Latin America citiesPLOS ONE

Dear Dr. Paiva,

Thank you for submitting your manuscript to PLOS ONE. After careful consideration, we feel that it has merit but does not fully meet PLOS ONE’s publication criteria as it currently stands. Therefore, we invite you to submit a revised version of the manuscript that addresses the points raised during the review process.

ACADEMIC EDITOR: The reviewer(s) have recommended publication, but also suggest some revisions to your manuscript. The main issue in this paper is about the theoretical contribution of this paper although methodologically and empirically they are well done. There are other issues that are highlighted by each reviewer.

We look forward to receiving your revised manuscript.

Kind regards,

Gabriel Hoh Teck Ling, PhD

Academic Editor

PLOS ONE

Journal Requirements:

The Salud Urbana en América Latina (SALURBAL)/Urban Health in Latin America project was funded by the Wellcome Trust. CIDACS has support from the Wellcome Trust UK (202912/B/16/Z)Moniz, Fundação Oswaldo Cruz, Salvador, Bahia, Brazil. 

The Salud Urbana en América Latina (SALURBAL)/Urban Health in Latin America project is funded by the Wellcome Trust [205177/Z/16/Z]. 

However, funding information should not appear in the Acknowledgments section or other areas of your manuscript. We will only publish funding information present in the Funding Statement section of the online submission form. 

The Salud Urbana en América Latina (SALURBAL)/Urban Health in Latin America project was funded by the Wellcome Trust. CIDACS has support from the Wellcome Trust UK (202912/B/16/Z)Moniz, Fundação Oswaldo Cruz, Salvador, Bahia, Brazil. 

7. We note that Figure 1 in your submission contain [map/satellite] images which may be copyrighted. All PLOS content is published under the Creative Commons Attribution License (CC BY 4.0), which means that the manuscript, images, and Supporting Information files will be freely available online, and any third party is permitted to access, download, copy, distribute, and use these materials in any way, even commercially, with proper attribution. For these reasons, we cannot publish previously copyrighted maps or satellite images created using proprietary data, such as Google software (Google Maps, Street View, and Earth). For more information, see our copyright guidelines: http://journals.plos.org/plosone/s/licenses-and-copyright.

Reviewers' comments:

Reviewer's Responses to Questions

**Comments to the Author**

1. Is the manuscript technically sound, and do the data support the conclusions?

Reviewer #1: Yes

Reviewer #2: Yes

2. Has the statistical analysis been performed appropriately and rigorously? 

Reviewer #1: Yes

Reviewer #2: Yes

3. Have the authors made all data underlying the findings in their manuscript fully available?

Reviewer #1: Yes

Reviewer #2: Yes

4. Is the manuscript presented in an intelligible fashion and written in standard English?

Reviewer #1: Yes

Reviewer #2: Yes

5. Review Comments to the Author

Reviewer #1: 1) The manuscript is a straightforward application of the urban scaling framework --- with insufficient argumentation as to why urban form should exhibit scaling behavior. Vague invocations of cities as complex systems---a claim which is not at all informative---do not constitute an argument as to why urban form should conform to a scaling relationship.

2) There is an established argument (derivation) as to why urban infrastructure should exhibit scaling behavior---see Lobo et al (2020) in Urban Studies. The manuscript exhibits ignorance of recent work in urban scaling.

3) The authors refer to a non existing controversy regarding the importance of urban delineation for investigating scaling: cities as social networks embedded in physical space should exhibit scaling; but not administratively defined city entities.

4) The connection of the presented results and urban planning is superficial and does not reflect how scaling results can inform urban policy.

5) The study of urban scaling over time is a topic of active work which the authors do not invoke; see, for example, https://royalsocietypublishing.org/doi/pdf/10.1098/rsif.2019.0846.

Reviewer #2: The paper entitled ‘A Scaling Investigation of Urban Form Features of Latin America Cities’ examine the investigates the urban scaling behavior in Latin American cities. Particularly, it focuses on the relationship between the physical urban form features with the population size of the cities. In general, the paper contributes interesting findings from the Latin America region. The current manuscript has a few weaknesses.

(1) At the first paragraph of page 3 mentioned about L3 level of analysis. But it does not include in the current study? Meanwhile, 5th Row of the second paragraph of the Descriptive Statistics section – ‘Table 3 shows the obtained results using the three L1 city definitions’ is not consistent with the label appears in Table 3 (L1AD, L2, L1UX). Further, the last paragraph of Descriptive Statistics section, L1AM is mentioned without any proper definition. Therefore, I think it would be great to adopt a set of clear and consistent labels for the different level of urban scaling analysis. Perhaps, using an additional figure or table could illustrate the definition ease readers to read and comprehend.

(2) For Scaling Analysis section, the discussion of the result does not truly reflect what it appears from the Table 4 and 5. For an example, the following discussion does not exactly explain the Table 4 results.

‘We note, however, that the Administrative area score with L2 and L1AD definitions, as well as that for Total urban patches with L1AD were quite low (8%, 19%, and 16% respectively).’

6. PLOS authors have the option to publish the peer review history of their article (what does this mean?). If published, this will include your full peer review and any attached files.

Reviewer #1: No

Reviewer #2: **Yes: **Bor Tsong Teh

---

## [Author Response · Author response to Decision Letter 0]

25 Apr 2023

1. Please ensure that your manuscript meets PLOS ONE's style requirements, including those for file naming. The PLOS ONE style templates can be found at.

Response: The manuscript format was placed in the style required by the journal.

Response: We added a Data availability topic. The scripts are now in a public repository on Github: https://github.com/sanchobuendia/Scaling-urban-form

The Salud Urbana en América Latina (SALURBAL)/Urban Health in Latin America project was funded by the Wellcome Trust. CIDACS has support from the Wellcome Trust UK (202912/B/16/Z)Moniz, Fundação Oswaldo Cruz, Salvador, Bahia, Brazil. 

Response: We have fixed this. This reads now:

Funding

The Salud Urbana en América Latina (SALURBAL)/Urban Health in Latin America project was funded by the Wellcome Trust [205177/Z/16/Z]. CIDACS has support from the Wellcome Trust UK (202912/B/16/Z) Instituto Gonçalo Moniz, Fundação Oswaldo Cruz, Salvador, Bahia, Brazil. UB was also supported by Office of the Director of the National Institutes of Health under award number DP5OD026429. The funders had no role in study design, data collection and analysis, decision to publish, or preparation of the manuscript.

The Salud Urbana en América Latina (SALURBAL)/Urban Health in Latin America project is funded by the Wellcome Trust [205177/Z/16/Z]. 

However, funding information should not appear in the Acknowledgments section or other areas of your manuscript. We will only publish funding information present in the Funding Statement section of the online submission form. 

The Salud Urbana en América Latina (SALURBAL)/Urban Health in Latin America project was funded by the Wellcome Trust. CIDACS has support from the Wellcome Trust UK (202912/B/16/Z)Moniz, Fundação Oswaldo Cruz, Salvador, Bahia, Brazil. 

Response: This part has been deleted from the text.

Response: There are not ethical or legal restrictions

Response: Data availability The SALURBAL project welcomes queries from anyone interested in learning more about its dataset and potential access to data. To learn more about SALURBAL’s dataset, visit https://drexel.edu/lac/ or contact the project at salurbal@drexel.edu. Furthermore, we are creating an open data repository to which we are adding the script, and data will be available at https://data.lacurbanhealth.org

Response: we have moved the ethical statement to the methods section.

7. We note that Figure 1 in your submission contain [map/satellite] images which may be copyrighted. All PLOS content is published under the Creative Commons Attribution License (CC BY 4.0), which means that the manuscript, images, and Supporting Information files will be freely available online, and any third party is permitted to access, download, copy, distribute, and use these materials in any way, even commercially, with proper attribution. For these reasons, we cannot publish previously copyrighted maps or satellite images created using proprietary data, such as Google software (Google Maps, Street View, and Earth). For more information, see our copyright guidelines: http://journals.plos.org/plosone/s/licenses-and-copyright.

Response: The figure was created in free software using non-copyrighted material. Therefore, it does not need copyright. We only include the name of the software in the text. 

Reviewer 1: 

Response: We thank the reviewer for the comments, which we understand to be directed mainly to the text in the first section Introduction. In the answers to the formulated questions we indicate the specific actions and changes that were made.

1) The manuscript is a straightforward application of the urban scaling framework --- with insufficient argumentation as to why urban form should exhibit scaling behavior. Vague invocations of cities as complex systems---a claim which is not at all informative---do not constitute an argument as to why urban form should conform to a scaling relationship.

Response: The first paragraph now presents only a general landscape of the research on city growth phenomena, finalizing with the mention of the scaling analysis framework. The mention to theoretical support for scaling ideas was moved to the next paragraph. We hope this provides a better justification for this study.

2) There is an established argument (derivation) as to why urban infrastructure should exhibit scaling behavior---see Lobo et al (2020) in Urban Studies. The manuscript exhibits ignorance of recent work in urban scaling.

Response: Thank you for the suggestions. The second paragraph has been rewritten, including now a substantial discussion on the theoretical support for scaling approaches. The corresponding references are introduced, including the specific item suggested above. We also included a comment on temporal scaling analysis and added three new references, including Lobo et al. 

3) The authors refer to a non existing controversy regarding the importance of urban delineation for investigating scaling: cities as social networks embedded in physical space should exhibit scaling; but not administratively defined city entities.

Response: Others (e.g., Cottineau, Hatna, Arcaute and Batty, 2017 –see https://www.sciencedirect.com/science/article/pii/S0198971516300448 ) do suggest this is a controversy. It seems that there is disagreement on whether this is controversial. 

The mention to the indicated controversy was removed from paragraph 6. In the paragraph 9, when we describe the different city definitions available on the Salurbal data sets, we indicate we carried out scaling analysis based on cities urban extents as detected by satellite imagery as well as on admistrative definition. We briefly comment the expected results taking into account the reviewer’s remarks above. 

4) The connection of the presented results and urban planning is superficial and does not reflect how scaling results can inform urban policy.

Response: We seek to bring a more in-depth discussion on this subject in paragraph 4, indicating more clearly how the results of the scale analysis can help policy makers in making decisions to improve the quality of life of all populations and subgroups.

5) The study of urban scaling over time is a topic of active work which the authors do not invoke; see, for example, https://royalsocietypublishing.org/doi/pdf/10.1098/rsif.2019.0846.

Response: We thank the reviewer for the suggestions. As indicated in our answer to query 2, we added a brief comment on this relevant aspect in the new paragraph 7. 

Reviewer 2: 

The paper entitled ‘A Scaling Investigation of Urban Form Features of Latin America Cities’ examine the investigates the urban scaling behavior in Latin American cities. Particularly, it focuses on the relationship between the physical urban form features with the population size of the cities. In general, the paper contributes interesting findings from the Latin America region. The current manuscript has a few weaknesses.

(1) At the first paragraph of page 3 mentioned about L3 level of analysis. But it does not include in the current study? Meanwhile, 5th Row of the second paragraph of the Descriptive Statistics section – ‘Table 3 shows the obtained results using the three L1 city definitions’ is not consistent with the label appears in Table 3 (L1AD, L2, L1UX). Further, the last paragraph of Descriptive Statistics section, L1AM is mentioned without any proper definition. Therefore, I think it would be great to adopt a set of clear and consistent labels for the different level of urban scaling analysis. Perhaps, using an additional figure or table could illustrate the definition ease readers to read and comprehend.

Response: These inconsistencies were identified and we included an explanation about L3, changed the description of Table 3, dropped the L1AM term and improved the description of levels.

(2) For Scaling Analysis section, the discussion of the result does not truly reflect what it appears from the Table 4 and 5. For an example, the following discussion does not exactly explain the Table 4 results.

‘We note, however, that the Administrative area score with L2 and L1AD definitions, as well as that for Total urban patches with L1AD were quite low (8%, 19%, and 16% respectively).’

Response: We adjusted the interpretation of the results in Tables 4 and 5. First, the scaling parameters of Table 4 was analyzed in two separate paragraphs (second and third paragraphs of the section). Second, the interpretation of Table 5 also was developed in a new paragraph (sixth paragraph of section).

---

## [Decision Letter · Decision Letter 1]

17 Jul 2023

PONE-D-22-28772R1A scaling investigation of urban form features in Latin America citiesPLOS ONE

Dear Dr. Paiva,

Thank you for submitting your manuscript to PLOS ONE. After careful consideration, we feel that it has merit but does not fully meet PLOS ONE’s publication criteria as it currently stands. Therefore, we invite you to submit a revised version of the manuscript that addresses the points raised during the review process.

ACADEMIC EDITOR: Thank you for the revised paper; however, there are some minor issues that need to be addressed before it can be further considered for publication.

We look forward to receiving your revised manuscript.

Kind regards,

Gabriel Hoh Teck Ling, PhD

Academic Editor

PLOS ONE

Journal Requirements:

Reviewers' comments:

Reviewer's Responses to Questions

**Comments to the Author**

1. If the authors have adequately addressed your comments raised in a previous round of review and you feel that this manuscript is now acceptable for publication, you may indicate that here to bypass the “Comments to the Author” section, enter your conflict of interest statement in the “Confidential to Editor” section, and submit your "Accept" recommendation.

Reviewer #2: (No Response)

2. Is the manuscript technically sound, and do the data support the conclusions?

Reviewer #2: Yes

3. Has the statistical analysis been performed appropriately and rigorously? 

Reviewer #2: Yes

4. Have the authors made all data underlying the findings in their manuscript fully available?

Reviewer #2: Yes

5. Is the manuscript presented in an intelligible fashion and written in standard English?

Reviewer #2: Yes

6. Review Comments to the Author

Reviewer #2: In this version of the paper, I can see some improvements from the previous one. I would like to further highlights my comments as follow:

(1) At the Method section, the terminology of L1 and L2 is mentioned in the discussion. However, there is no proper introduction and definition about them in the first place.

(2) Table 3 contains L1AD density, L2 density and L1UX density. Is it necessary to display them? What is the purpose for having them? I do not find any discussion about these three items.

(3) For Scaling Analysis section, the discussion of the result does not truly reflect what it appears from the Table 4 and 5.

Do make sure the items (examples) below in Scaling Analysis section are consistent with information displayed in Table 4 and 5.

- ‘The same occurs for for Number of urban patches in L1AD, with 15.8%’

- ‘The Large road variables has an exponent smaller than but very close to 1 in the L2 and L1UX levels.’

(4) The current Discussion section mainly focuses on the technical discussion – power scaling result, statistical result, and description result, as well as the result relation to other studies. Nonetheless, little discussion on the idea of urban planning is to explain the scaling behavior. Further, the current discussion misses to establish an important connection between the research findings and its implication for better urban planning in Latin America cities.

7. PLOS authors have the option to publish the peer review history of their article (what does this mean?). If published, this will include your full peer review and any attached files.

Reviewer #2: No

---

## [Author Response · Author response to Decision Letter 1]

2 Aug 2023

Dear reviewer, we thank you for all your comments and suggestions aimed at improving our article. Below are the answers.

6. Review Comments to the Author

Reviewer #2: In this version of the paper, I can see some improvements from the previous one. I would like to further highlights my comments as follow:

(1) At the Method section, the terminology of L1 and L2 is mentioned in the discussion. However, there is no proper introduction and definition about them in the first place.

Response: Thanks for this remark. We removed any reference about L1 and L2 in the Method section. However, we note that the definition of “L1” and “L2” is available in the Data section together with the reference to the seminal project protocol article, as follows: “Being more specific, we restrain our study to urban agglomerations classified as “cities” at the L1 and L2 SALURBAL levels [36], which amounts to taking n=371 or 1436 at the L1 or L2 levels, respectively”. (Quistberg et al. Building a data platform for cross-country urban health studies: the SALURBAL study. Journal of urban health. 2019 Apr 15;96:311-37).

(2) Table 3 contains L1AD density, L2 density and L1UX density. Is it necessary to display them? What is the purpose for having them? I do not find any discussion about these three items.

Response: Thanks for this comment. We reformatted Table 3 to present only the results at the L1AD, L2 and L1UX level. The discussion of the items encompasses the paragraphs requested by the reviewer’s comment (4) on questions about urban planning. Please see specific answer to comment 4 below. 

(3) For Scaling Analysis section, the discussion of the result does not truly reflect what it appears from the Table 4 and 5.

Do make sure the items (examples) below in Scaling Analysis section are consistent with information displayed in Table 4 and 5.

- ‘The same occurs for Number of urban patches in L1AD, with 15.8%’

Thanks for highlighting this. Fixed for L2. “The same occurs for Number of urban patches in L2, with 15.8%.”

- ‘The Large Road variables has an exponent smaller than but very close to 1 in the L2 and L1UX levels.’

Thanks for that. It is fixed: “The large road variable has an exponent β very close to 1 in the L1AD and L1UX levels”.

(4) The current Discussion section mainly focuses on the technical discussion – power scaling result, statistical result, and description result, as well as the result relation to other studies. Nonetheless, little discussion on the idea of urban planning is to explain the scaling behavior. Further, the current discussion misses to establish an important connection between the research findings and its implication for better urban planning in Latin America cities.

Response: Thanks for this comment. As mentioned, we inserted the following sentences in the discussion section: 

“The results found here will serve as a basis for comparison with further studies targeted to similar analyses for cities in other regions. We emphasize the importance of using appropriate city definitions in scaling analyses. Our study reveals that city definitions based on the actual extent of the urban area yield more accurate insights compared to administrative definitions. This highlights the need for urban planners in Latin America to consider the physical boundaries of urban areas when making decisions about resource allocation, spatial organization, and infrastructure development. By understanding the scaling behavior of built environment variables in relation to the actual urban extent, planners can create more effective strategies to address the specific needs and challenges of rapidly expanding cities.

Furthermore, the study's discovery of a degree of universality in the relationship between population size and built environment variables across diverse Latin American cities has important implications for urban planning. Despite varying economic, historical, and cultural differences, the fundamental mechanisms that drive urban growth appear to exhibit common patterns. This finding suggests that certain strategies for city development may be applicable across different cities in the region, providing valuable guidance for planners seeking sustainable and efficient urbanization approaches.”

---

## [Decision Letter · Decision Letter 2]

16 Oct 2023

A scaling investigation of urban form features in Latin America cities

PONE-D-22-28772R2

Dear Dr. Paiva,

We’re pleased to inform you that your manuscript has been judged scientifically suitable for publication and will be formally accepted for publication once it meets all outstanding technical requirements.

Kind regards,

Gábor Vattay, PhD, DSc

Academic Editor

PLOS ONE

Additional Editor Comments (optional):

Reviewers' comments:

Reviewer's Responses to Questions

**Comments to the Author**

1. If the authors have adequately addressed your comments raised in a previous round of review and you feel that this manuscript is now acceptable for publication, you may indicate that here to bypass the “Comments to the Author” section, enter your conflict of interest statement in the “Confidential to Editor” section, and submit your "Accept" recommendation.

Reviewer #2: All comments have been addressed

Reviewer #3: (No Response)

2. Is the manuscript technically sound, and do the data support the conclusions?

Reviewer #2: Yes

Reviewer #3: Yes

3. Has the statistical analysis been performed appropriately and rigorously? 

Reviewer #2: Yes

Reviewer #3: Yes

4. Have the authors made all data underlying the findings in their manuscript fully available?

Reviewer #2: Yes

Reviewer #3: (No Response)

5. Is the manuscript presented in an intelligible fashion and written in standard English?

Reviewer #2: Yes

Reviewer #3: (No Response)

6. Review Comments to the Author

Reviewer #2: (No Response)

Reviewer #3: This is my first time reviewing the manuscript. In general, I think it is well done and the authors have mostly addressed the previous reviews sufficiently. I will encourage the authors to do three things below. Items 1 and 2 can be addressed in the discussion. Point 3 can be included in the supplemental.

1) The authors compared Brazil to other Latin American countries and this is justified because of differences in colonization history. However, the authors do not provide justification for lumping all of the LA countries together. I raise this because recent studies using global databases found mixed results among Latin American countries for the scaling of area and population (Ortman et al. 2020 PLoS one; Burger et al. 2022 Front Con Sci). I realize the authors are restricted to only a few countries with sufficient sample sizes to interpret the within country scaling exponent. This may be beyond the scope of this study, but can the authors justify (in methods or discussion) why they wouldn't expect variation among countries to driver their patterns, for example if Mexico was largely driving the relationship?

2) Interpretation of other variables may be enhanced by understanding how density varies across countries of different sizes. In your analysis and as you briefly summarize, it appears density increases with city size, yet this is in contrast to above studies of the region. I would like the authors to expand the discussion section on density in relation to the above studies and how it might influence (or be influenced by) the other urban attributes. This may be combined with a discussion of the above on how combining all LA countries may obscure among country variation. Related to the other reviewer's comments, this may be especially important for implications for urban planners that would likely work at country-level or even smaller scales.

3) The R2 values are variable among the urban attributes analyzed, which the author's do addressed. It would be useful to see the scatterplots of all of the variables analyzed in the supplemental so that the reader can see how the residual variation occurs.

7. PLOS authors have the option to publish the peer review history of their article (what does this mean?). If published, this will include your full peer review and any attached files.

Reviewer #2: No

Reviewer #3: **Yes: **Joseph R Burger

---

## [Editor Report · Acceptance letter]

18 Oct 2023

PONE-D-22-28772R2 

A scaling investigation of urban form features in Latin America cities 

Dear Dr. Paiva:

I'm pleased to inform you that your manuscript has been deemed suitable for publication in PLOS ONE. Congratulations! Your manuscript is now with our production department. 

Kind regards, 

on behalf of

Dr. Gábor Vattay 

Academic Editor

PLOS ONE